# Unsupervised Meta-Learning for Few-Shot Image Classification

**Siavash Khodadadeh, Ladislau Bölöni**
Dept. of Computer Science
University of Central Florida
`siavash.khodadadeh@knights.ucf.edu, lboloni@cs.ucf.edu`

**Mubarak Shah**
Center for Research in Computer Vision
University of Central Florida
`shah@crcv.ucf.edu`

## Abstract

Few-shot or one-shot learning of classifiers requires a significant inductive bias towards the type of task to be learned. One way to acquire this is by meta-learning on tasks similar to the target task. In this paper, we propose UMTRA, an algorithm that performs unsupervised, model-agnostic meta-learning for classification tasks.

The meta-learning step of UMTRA is performed on a flat collection of unlabeled images. While we assume that these images can be grouped into a diverse set of classes and are relevant to the target task, no explicit information about the classes or any labels are needed. UMTRA uses random sampling and augmentation to create synthetic training tasks for meta-learning phase. Labels are only needed at the final target task learning step, and they can be as little as one sample per class.

On the Omniglot and Mini-Imagenet few-shot learning benchmarks, UMTRA outperforms every tested approach based on unsupervised learning of representations, while alternating for the best performance with the recent CACTUs algorithm. Compared to supervised model-agnostic meta-learning approaches, UMTRA trades off some classification accuracy for a reduction in the required labels of several orders of magnitude.

## 1 Introduction

Meta-learning or "learning-to-learn" approaches have been proposed in the neural networks literature since the 1980s [29, 4]. The general idea is to prepare the network through several learning tasks $\mathcal{T}_1 \ldots \mathcal{T}_n$, in a *meta-learning phase* such that when presented with the target task $\mathcal{T}_{n+1}$, the network will be ready to learn it as efficiently as possible.

Recently proposed model-agnostic meta-learning approaches [11, 23] can be applied to any differentiable network. When used for classification, the target learning phase consists of several gradient descent steps on a backpropagated supervised classification loss. Unfortunately, these approaches require the learning tasks $\mathcal{T}_i$ to have the same supervised learning format as the target task. Acquiring labeled data for a large number of tasks is not only a problem of cost and convenience but also puts conceptual limits on the type of problems that can be solved through meta-learning. If we need to have labeled training data for tasks $\mathcal{T}_1 \ldots \mathcal{T}_n$ in order to learn task $\mathcal{T}_{n+1}$, this limits us to task types that are variations of tasks known and solved (at least by humans).

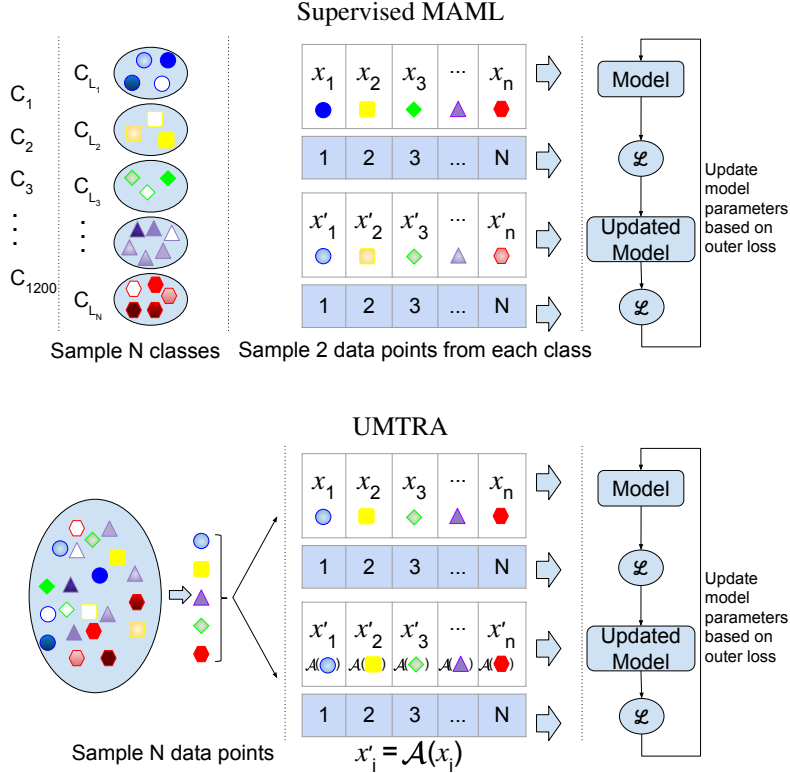

Figure 1: The process of creation of the training and validation data of the meta-training task $\mathcal{T}$. (top) Supervised MAML: We start from a dataset where the samples are labeled with their class. The training data is created by sampling $N$ distinct classes $C_{L_i}$, and choosing a random sample $x_i$ from each. The validation data is created by choosing a different sample $x_i'$ from the same class. (bottom) UMTRA: We start from a dataset of unlabeled data. The training data is created by randomly choosing N samples $x_i$ from the dataset. The validation data is created by applying the augmentation function $\mathcal{A}$ to each sample from the training data. For both MAML and UMTRA, artificial temporary labels $1, 2 \ldots N$ are used.

In this paper, we propose an algorithm called Unsupervised Meta-learning with Tasks constructed by Random sampling and Augmentation (UMTRA) that performs meta-learning of one-shot or few-shot classifiers in an unsupervised manner on an unlabeled dataset. Instead of starting from a collection of labeled tasks, $\{\ldots \mathcal{T}_i \ldots\}$, UMTRA starts with a collection of unlabeled data $\mathcal{U} = \{\ldots x_i \ldots\}$. We have only a set of relatively easy-to-satisfy requirements towards $\mathcal{U}$: Its objects have to be drawn from the same distribution as the objects classified in the target task and it must have a set of classes significantly larger than the number of classes of the final classifier. Starting from this unlabeled dataset, UMTRA uses statistical diversity properties and domain specific augmentations to generate the training and validation data for a collection of synthetic tasks, $\{\ldots \mathcal{T}_i' \ldots\}$. These tasks are then used in the meta-learning process based on a modified classification variant of the MAML algorithm [11]. Figure 1 summarizes the differences between the original supervised MAML model and the process of generating synthetic tasks from unsupervised data in UMTRA.

The contributions of this paper can be summarized as follows:

- We describe a novel algorithm that allows unsupervised, model-agnostic meta-learning for few-shot classification by generating synthetic meta-learning data with artificial labels.
- From a theoretical point of view, we demonstrate a relationship between generalization error and the loss backpropagated from the validation set in MAML. Our intuition is that we can generate unsupervised validation tasks which can perform effectively if we are able to span the space of the classes by generating useful samples with augmentation.
- On all the Omniglot and Mini-Imagenet few-shot learning benchmarks, UMTRA outperforms every tested approach based on unsupervised learning of representations. It also

achieves a significant percentage of the accuracy of the supervised MAML approach, while requiring vastly fewer labels. For instance, for 5-way 5-shot classification on the Omniglot dataset UMTRA obtains a 95.43% accuracy with only **25** labels, while supervised MAML obtains 98.83% with 24025. Compared with recent unsupervised meta-learning approaches building on top of stock MAML, UMTRA alternates for the best performance with the CACTUs algorithm.

## 2 Related Work

Few-shot or one-shot learning of classifiers has significant practical applications. Unfortunately, the few-shot learning model is not a good fit to the traditional training approaches of deep neural networks, which work best with large amounts of data. In recent years, significant research targeted approaches to allow deep neural networks to work in few-shot learning settings. One possibility is to perform transfer learning, but it was found that the accuracy decreases if the target task diverges from the trained task. One solution to mitigate this is through the use of an adversarial loss [18].

A large class of approaches aim to enable few-shot learning by *meta-learning* - the general idea being that the meta-learning prepares the network to learn from the small amount of training data available in the few-shot learning setting. Note that meta-learning can be also used in other computer vision applications, such as fast adaptation for tracking in video [25]. The mechanisms through which meta-learning is implemented can be loosely classified in two groups. One class of approaches use a custom network architecture for encoding the information acquired during the meta-learning phase, for instance in fast weights [3], neural plasticity values [21], custom update rules [20], the state of temporal convolutions [22] or in the memory of an LSTM [27]. The advantage of this approach is that it allows us to fine-tune the architecture for the efficient encoding of the meta-learning information. A disadvantage, however, is that it constrains the type of network architectures we can use; innovations in network architectures do not automatically transfer into the meta-learning approach. In a custom network architecture meta-learning model, the target learning phase is not the customary network learning, as it needs to take advantage of the custom encoding.

A second, model-agnostic class of approaches aim to be usable for any differentiable network architecture. Examples of these algorithms are MAML [11] or Reptile [23], where the aim is to encode the meta-learning in the weights of the network, such that the network performs the target learning phase with efficient gradients. Approaches that customize the learning rates [19] during meta-training can also be grouped in this class. For this type of approaches, the target learning phase uses the well-established learning algorithms that would be used if learning from scratch (albeit it might use specific hyperparameter settings, such as higher learning rates). We need to point out, however, that the meta-learning phase uses custom algorithms in these approaches as well (although they might use the standard learning algorithm in the inner loop, such as in the case of MAML). A recent work similar in spirit to ours is the CACTUs unsupervised meta-learning model described in [14].

In this paper, we perform unsupervised meta-learning. Our approach generates tasks from unlabeled data which will help it to understand the structures of the relevant supervised tasks in the future. One should note that these relevant supervised tasks in the future do not have any intersection with the tasks which are used during the meta-learning. For instance, Wu *et al*. perform unsupervised learning by recognizing a certain internal structure between dataset classes [32]. By learning this structure, the approach can be extended to semi-supervised learning. In addition, Pathak *et al*. propose a method which learns object features in an interesting unsupervised way by detecting movement patterns of segmented objects [26]. These approaches are orthogonal to ours. We do not make assumptions that the unsupervised data shares classes with the target learning (in fact, we explicitly forbid it). Finally, [13] define unsupervised meta-learning in reinforcement learning context. The authors study how to generate tasks with synthetic reward functions (without supervision) such that when the policy network is meta trained on them, they can learn real tasks with manually defined reward functions (with supervision) much more quickly and with fewer samples.

# 3 The UMTRA algorithm

## 3.1 Preliminaries

We consider the task of classifying samples $\mathbf{x}$ drawn from a domain $\mathbf{X}$ into classes $\mathbf{y}_i \in Y = \{C_1, \ldots, C_N\}$. The classes are encoded as one-hot vectors of dimensionality $N$. We are interested in learning a classifier $f_\theta$ that outputs a probability distribution over the classes. It is common to envision $f$ as a deep neural network parameterized by $\theta$, although this is not the only possible choice.

We package a certain supervised learning task, $\mathcal{T}$, of type $(N, K)$, that is with $N$ classes of $K$ training samples each, as follows. The training data will have the form $(x_i, y_i)$, where $i = 1 \ldots N \times K$, $\mathbf{x}_i \in X$ and $\mathbf{y}_i \in Y$, with exactly $K$ samples for each value of $y_i$. In the recent meta-learning literature, it is often assumed that the task $\mathcal{T}$ has $K$ samples of each class for training *and* (separately), $K$ samples for validation $(x_j^v, y_j^v)$.

In supervised meta-learning, we have access to a collection of tasks $\mathcal{T}_1 \ldots \mathcal{T}_n$ drawn from a specific distribution, with both supervised training and validation data. The meta-learning phase uses this collection of tasks, while the target learning uses a new task $\mathcal{T}$ with supervised learning data but no validation data.

## 3.2 Model

Unsupervised meta-learning retains the goal of meta-learning by preparing a learning system for the rapid learning of the target task $\mathcal{T}$. However, instead of the collection of tasks $\mathcal{T}_1 \ldots \mathcal{T}_n$ and their associated labeled training data, we only have an unlabeled dataset $\mathcal{U} = \{\ldots x_i \ldots\}$, with samples drawn from the same distribution as the target task. We assume that every element of this dataset is associated with a natural class $C_1 \ldots C_c$, $\forall x_i \exists j$ such that $x_i \in C_j$. We will assume that $N \ll c$, that is, the number of natural classes in the unsupervised dataset is much higher than the number of classes in the target task. These requirements are much easier to satisfy than the construction of the tasks for supervised meta-learning - for instance, simply stripping the labels from datasets such as Omniglot and Mini-ImageNet satisfies them.

The pseudo-code of the UMTRA algorithm is described in Algorithm 1. In the following, we describe the various parts of the algorithm in detail. In order to be able to run the UMTRA algorithm on unsupervised data, we need to create tasks $\mathcal{T}_i$ from the unsupervised data that can serve the same role as the meta-learning tasks serve in the full MAML algorithm. For such a task, we need to create both the training data $\mathcal{D}$ and the validation data $\mathcal{D}'$.

**Creating the training data:** In the original form of the MAML algorithm, the training data of the task $\mathcal{T}$ must have the form $(x, y)$, and we need $N \times K$ of them. The exact labels used during the meta-training step are not relevant, as they are discarded during the meta-training phase. They can be thus replaced with artificial labels, by setting them $y \in \{1, \ldots N\}$. It is however, important that the labels maintain class distinctions: if two data points have the same label, they should also have the same artificial labels, while if they have different labels, they should have different artificial labels.

The first difference between UMTRA and MAML is that during the meta-training phases, we always perform one-shot learning, with $K = 1$. Note that during the target learning phase we can still set values of $K$ different from 1. The training data is created as the set $\mathcal{D}_i = \{(x_1, 1), \ldots (x_N, N)\}$, with $x_i$ *sampled randomly* from $\mathcal{U}$.

Let us see how this training data construction satisfy the class distinction conditions. The first condition is satisfied because there is only one sample for each label. The second condition is satisfied statistically by the fact that $N \ll c$, where $c$ is the total number of classes in the dataset. If the number of samples is significantly smaller than the number of classes, it is likely that all the samples will be drawn from different classes. If we assume that the samples are equally distributed among the classes (e.g. $m$ samples for each class), the probability that all samples are in a different class is equal to

$$P = \frac{(c \cdot m) \cdot ((c-1) \cdot m)...((c-N+1) \cdot m)}{(c \cdot m) \cdot (c \cdot m - 1)...(c \cdot m - N + 1)} = \frac{c! \cdot m^N \cdot (c \cdot m - N)!}{(c-N)! \cdot (c \cdot m)!} \tag{1}$$

To illustrate this, the probability for 5-way classification on the Omniglot dataset used with each of the 1200 characters is a separate class ($c = 1200$, $N = 5$) is 99.21%. For Mini-ImageNet ($c = 64$), the probability is 85.23%, while for the full ImageNet it would be about 99%.

**Algorithm 1:** Unsupervised Meta-learning with Tasks constructed by Random sampling and Augmentation (UMTRA)

---

**require :** $N$: class-count, $N_{MB}$: meta-batch size, $N_U$: no. of updates
**require :** $\mathcal{U} = \{\ldots x_i \ldots\}$ unlabeled dataset
**require :** $\alpha$, $\beta$: step size hyperparameters
**require :** $\mathcal{A}$: augmentation function

1  randomly initialize $\theta$;
2  **while** *not done* **do**
3      **for** *i in* $1 \ldots N_{MB}$ **do**
4          Sample $N$ data points $x_1 \ldots x_N$ from $\mathcal{U}$;
5          $\mathcal{T}_i \leftarrow \{x_1, \ldots x_N\}$;
6      **end**
7      **foreach** $\mathcal{T}_i$ **do**
8          Generate training set $\mathcal{D}_i = \{(x_1, 1), \ldots, (x_N, N)\}$;
9          $\theta'_i = \theta$;
10         **for** *j in* $1 \ldots N_U$ **do**
11             Evaluate $\nabla_{\theta'_i} \mathcal{L}_{\mathcal{T}_i}(f_{\theta'_i})$;
12             Compute adapted parameters with gradient descent: $\theta'_i = \theta'_i - \alpha \nabla_{\theta'_i} \mathcal{L}_{\mathcal{T}_i}(f_{\theta'_i})$;
13         **end**
14         Generate validation set for the meta-update $\mathcal{D}'_i = \{(\mathcal{A}(x_1), 1), \ldots, (\mathcal{A}(x_N), N)\}$
15     **end**
16     Update $\theta \leftarrow \theta - \beta \nabla_\theta \sum_{\mathcal{T}_i} \mathcal{L}_{\mathcal{T}_i}(f_{\theta'_i})$ using each $\mathcal{D}'_i$;
17 **end**

---

**Creating the validation data:** For the MAML approach, the validation data of the meta-training tasks is actually training data in the outer loop. It is thus required that we create a validation dataset $\mathcal{D}'_i = \{(x'_1, 1), \ldots (x'_N, N)\}$ for each task $\mathcal{T}_i$. Thus we need to create appropriate validation data for the synthetic task. A minimum requirement for the validation data is to be correctly labeled in the given context. This means that the synthetic numerical label should map in both cases to the same class in the unlabeled dataset: $\nexists C$ such that $x_i, x'_i \in C$.

In the original MAML model, these $x'_i$ values are labeled examples part of the supervised dataset. In our case, picking such $x'_i$ values is non-trivial, as we don't have access to the actual class. Instead, we propose to *create* such a sample by augmenting the sample used in the training data using an *augmentation function* $x'_i = \mathcal{A}(x_i)$ which is a hyperparameter of the UMTRA algorithm. A requirement towards the augmentation function is to maintain class membership $x \in C \Rightarrow \mathcal{A}(x) \in C$. We should aim to construct the augmentation function to verify this property for the given dataset $\mathcal{U}$, based on what we know about the domain described by the dataset. However, as we do not have access to the classes, such a verification is not practically possible on a concrete dataset.

Another choice for the augmentation function $\mathcal{A}$ is to apply some kind of domain-specific change to the images or videos. Examples of these include setting some of the pixel values to zero in the image (Figure 2, left), or translating the pixels of the training image by some amount (eg. between -6 and 6).

The overall process of generating the training data from the unlabeled dataset in UMTRA and the differences from the supervised MAML approach is illustrated in Figure 1.

### 3.3 Some theoretical considerations

While a full formal model of the learning ability of the UMTRA algorithm is beyond the scope of this paper, we can investigate some aspects of its behavior that shed light into why the algorithm is working, and why augmentation improves its performance. Let us denote our network with a parameterized function $f_\theta$. As we want to learn a few-shot classification task, $\mathcal{T}$ we are searching for the corresponding function $f_\mathcal{T}$, to which we do not have access. To learn this function, we use the training dataset, $D_\mathcal{T} = \{(x_i, y_i)\}_{i=1}^{n \times k}$. For this particular task, we update our parameters (to $\theta'$) to fit this task's training dataset. In other words, we want $f_{\theta'}$ to be a good approximation of

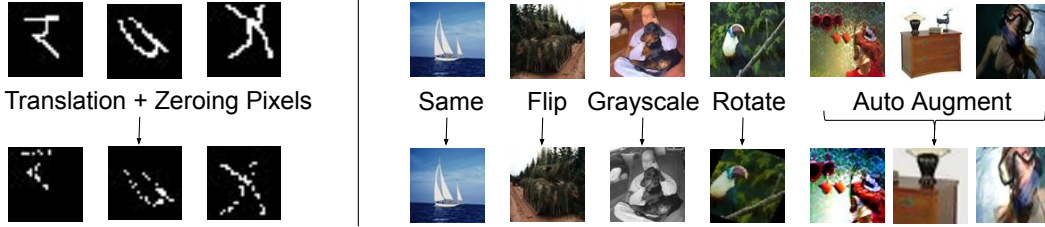

Figure 2: Augmentation techniques on Omniglot (left) and Mini-Imagenet (right). Top row: Original images in training data. Bottom: augmented images for the validation set, transformed with an augmentation function $\mathcal{A}$. Auto Augment [8] applies augmentations from a learned policy based on combinations of translation, rotation, or shearing.

$f_{\mathcal{T}}$. Finding $\theta'$ such that, $\theta' = \underset{\theta}{\operatorname{argmin}} \sum_{(x_i, y_i) \in D_{\mathcal{T}}} \mathcal{L}(y_i, f_\theta(x_i))$ is ill-defined because there are more than one solution for it. In meta-learning, we search for the $\theta'$ value that gives us the minimum generalization error, the measure of how accurately an algorithm is able to predict outcome values for unseen data [1]. We can estimate the generalization error based on sampled data points from the same task. Without loss of generality, let us consider a sampled data point $(x_0, y_0)$. We can estimate generalization error on this point as $\mathcal{L}(y_0, f_{\theta'}(x_0))$. In case of mean squared error, and by accepting irreducible error $\epsilon \sim \mathcal{N}(0, \sigma)$, we can decompose the expected generalization error as follows [16, 12]:

$$E\left[\mathcal{L}(y_0, f_{\theta'}(x_0))\right] = \left(E[f_{\theta'}(x_0)] - f_{\mathcal{T}}(x_0)\right)^2 + E\left[\left(f_{\theta'}(x_0)\right)^2\right] - E\left[f_{\theta'}(x_0)\right]^2 + \sigma^2 \quad (2)$$

In this equation, when $(x_0, y_0) \notin D_{\mathcal{T}}$ we have $E[(f_{\theta'}(x_0))^2] - E[f_{\theta'}(x_0)]^2 = 0$, which means that the estimation of the generalization error on these samples will be as unbiased as possible (only biased by $\sigma^2$). On the other hand, if $(x_0, y_0) \in D_{\mathcal{T}}$, the estimation of the error is going to be highly biased. We conjecture that similar results will be observed for other loss functions as well with the estimate of the loss function being more biased if the samples are from the training data rather than outside it. As in the outer loop of MAML estimates the generalization error on a validation set for each task in a batch of tasks, it is important to keep the validation set separate from the training set, as this estimate will be eventually applied to the starter network.

In contrast, if we pick our validation set as points in $D_{\mathcal{T}}$, our algorithm is going to learn to minimize a biased estimation of the generalization error. Our experiments also show that if we choose the same data for train and test ($\mathcal{A}(x) = x$), we will end up with an accuracy almost the same as training from scratch. UMTRA, however, tries to improve the estimation of generalization error with augmentation techniques. Our experiments show that by applying UMTRA with good choice of function for augmentation, we can achieve comparable results with supervised meta-learning algorithms. In our supplementary material, we show that UMTRA is able to adapt very quickly with just few iterations to a new task. Last but not least, in comparison with CACTUs algorithm which applies advanced clustering algorithms such as DeepCluster [6], ACAI [5], and BiGAN [10] to generate train and validation set for each task, our method does not require clustering.

## 4 Experiments

### 4.1 UMTRA on the Omniglot dataset

Omniglot [17] is a dataset of handwritten characters frequently used to compare few-shot learning algorithms. It comprises 1623 characters from 50 different alphabets. Every character in Omniglot has 20 different instances each was written by a different person. To allow comparisons with other published results, in our experiments we follow the experimental protocol described in [28]: 1200 characters were used for training, 100 characters were used for validation and 323 characters were used for testing.

UMTRA, like the supervised MAML algorithm, is model-agnostic, that is, it does not impose conditions on the actual network architecture used in the learning. This does not, of course, mean that

Table 1: The influence of augmentation function on the accuracy of UMTRA for 5-way one-shot classification on the (Left: Omniglot dataset, Right: Mini-Imagenet dataset). For all cases, we use meta-batch size $N_{MB} = 4$ and number of updates $N_U = 5$, except the ones with best hyperparameters.

| Augmentation Function $\mathcal{A}$ | Accuracy | Augmentation Function $\mathcal{A}$ | Accuracy |
|---|---|---|---|
| *Training from scratch* | 52.50 | *Training from scratch* | 24.17 |
| $\mathcal{A} = \mathbb{1}$ | 52.93 | $\mathcal{A} = \mathbb{1}$ | 26.49 |
| $\mathcal{A}$ = randomly zeroed pixels | 56.23 | $\mathcal{A}$ = Shift + random flip | 30.16 |
| $\mathcal{A}$ = randomly zeroed pixels (with best hyperparameters) | 67.00 | $\mathcal{A}$ = Shift + random flip + randomly change to grayscale | 32.80 |
| $\mathcal{A}$ = randomly zeroed pixels + random shift (with best hyperparameters) | **83.80** | $\mathcal{A}$ = Shift + random flip + random rotation + color distortions | 35.09 |
|  |  | $\mathcal{A}$ = Auto Augment [8] | **39.93** |
| *Supervised MAML* | 98.7 | *Supervised MAML* | 46.81 |

Table 2: The effect of hyperparameters meta-batch size, $N_{MB}$, and number of updates, $N_U$ on accuracy. Omniglot 5-way one shot.

| # Updates \ $N_{MB}$ | 1 | 2 | 4 | 8 | 16 | 25 |
|---|---|---|---|---|---|---|
| 1 | 67.08 | 79.04 | 80.72 | 81.60 | 82.72 | **83.80** |
| 5 | 76.08 | 76.68 | 77.20 | 79.56 | 81.12 | 83.32 |
| 10 | 79.20 | 79.24 | 80.92 | 80.68 | 83.52 | 83.26 |

the algorithm performs identically for every network structure and dataset. In order to separate the performance of the architecture and the meta-learner, we run our experiments using an architecture originally proposed in [31]. This classifier uses four 3 x 3 convolutional modules with 64 filters each, followed by batch normalization [15], a ReLU nonlinearity and 2 x 2 max-pooling. On the resulting feature embedding, the classifier is implemented as a fully connected layer followed by a softmax layer.

UMTRA has a relatively large hyperparameter space that includes the augmentation function. As pointed out in a recent study involving performance comparisons in semi-supervised systems [24], excessive tuning of hyperparameters can easily lead to an overestimation of the performance of an approach compared to simpler approaches. Thus, for the comparison in the remainder of this paper, we keep a relatively small budget for hyperparameter search: beyond basic sanity checks, we only tested 5-10 hyperparameter combinations per dataset, without specializing them to the N or K parameters of the target task. Table 1, left, shows several choices for the augmentation function for the 5-way one-shot classification on Omniglot. Based on this table, in comparing with other approaches, we use an augmentation function consisting of randomly zeroed pixels and random shift.

In our experiments, we realized two of the most important hyperparameters in meta-learning are meta-batch size, $N_{MB}$, and number of updates, $N_U$. In table 2, we study the effects of these hyperparameters on the accuracy of the network for the randomly zeroed pixels and random shift augmentation. Based on this experiment, we decide to fix the meta-batch size to 25 and number of updates to 1.

In order to find out the relationship between the level of the augmentation and accuracy, we apply different levels of augmentation on images. If the generated samples are different from current observation but within the same class manifold, UMTRA performs well. The results of this experiment are shown in table 3.

The second consideration is what sort of baseline we should use when evaluating our approach on a few-shot learning task? Clearly, supervised meta-learning approaches such as an original MAML [11]

Table 3: The effect of the augmentation level on UMTRA's accuracy on the Omniglot dataset. In all of the experiments we use random pixel zeroing with meta-batch size $N_{MB} = 25$ and number of updates $N_U = 1$.

| Translation Range (Pixels) | 0 | 0-3 | 3-6 | 0-6 | 6-9 | 9-12 | 0-9 |
|---|---|---|---|---|---|---|---|
| **Accuracy %** | 67.0 | 82.8 | 80.4 | **83.8** | 79.8 | 77 | 80.4 |

Table 4: Accuracy in % of N-way K-shot (N,K) learning methods on the Omniglot and Mini-Imagenet datasets. The ACAI / DC label means ACAI Clustering on Omniglot and DeepCluster on Mini-Imagenet. The source of non-UMTRA values is [14].

| Algorithm (N, K) | Clustering | Omniglot | | | | Mini-Imagenet | | | |
|---|---|---|---|---|---|---|---|---|---|
| | | (5,1) | (5,5) | (20,1) | (20,5) | (5,1) | (5,5) | (5,20) | (5,50) |
| *Training from scratch* | N/A | 52.50 | 74.78 | 24.91 | 47.62 | 27.59 | 38.48 | 51.53 | 59.63 |
| $k_{nn}$-nearest neighbors | BiGAN | 49.55 | 68.06 | 27.37 | 46.70 | 25.56 | 31.10 | 37.31 | 43.60 |
| linear classifier | BiGAN | 48.28 | 68.72 | 27.80 | 45.82 | 27.08 | 33.91 | 44.00 | 50.41 |
| MLP with dropout | BiGAN | 40.54 | 62.56 | 19.92 | 40.71 | 22.91 | 29.06 | 40.06 | 48.36 |
| cluster matching | BiGAN | 43.96 | 58.62 | 21.54 | 31.06 | 24.63 | 29.49 | 33.89 | 36.13 |
| CACTUs-MAML | BiGAN | 58.18 | 78.66 | 35.56 | 58.62 | 36.24 | 51.28 | 61.33 | 66.91 |
| CACTUs-ProtoNets | BiGAN | 54.74 | 71.69 | 33.40 | 50.62 | 36.62 | 50.16 | 59.56 | 63.27 |
| $k_{nn}$-nearest neighbors | ACAI / DC | 57.46 | 81.16 | 39.73 | 66.38 | 28.90 | 42.25 | 56.44 | 63.90 |
| linear classifier | ACAI / DC | 61.08 | 81.82 | 43.20 | 66.33 | 29.44 | 39.79 | 56.19 | 65.28 |
| MLP with dropout | ACAI / DC | 51.95 | 77.20 | 30.65 | 58.62 | 29.03 | 39.67 | 52.71 | 60.95 |
| cluster matching | ACAI / DC | 54.94 | 71.09 | 32.19 | 45.93 | 22.20 | 23.50 | 24.97 | 26.87 |
| CACTUs-MAML | ACAI / DC | 68.84 | 87.78 | 48.09 | 73.36 | 39.90 | **53.97** | **63.84** | **69.64** |
| CACTUs-ProtoNets | ACAI / DC | 68.12 | 83.58 | 47.75 | 66.27 | 39.18 | 53.36 | 61.54 | 63.55 |
| UMTRA (ours) | N/A | **83.80** | **95.43** | **74.25** | **92.12** | **39.93** | 50.73 | 61.11 | 67.15 |
| *MAML (Supervised)* | N/A | 94.46 | 98.83 | 84.60 | 96.29 | 46.81 | 62.13 | 71.03 | 75.54 |
| *ProtoNets (Supervised)* | N/A | 98.35 | 99.58 | 95.31 | 98.81 | 46.56 | 62.29 | 70.05 | 72.04 |

are expected to outperform our approach, as they use a labeled training set. A simple baseline is to use the same network architecture being trained from scratch with only the final few-shot labeled set. If our algorithm takes advantage of the unsupervised training set $\mathcal{U}$, as expected, it should outperform this baseline.

A more competitive comparison can be made against networks that are first trained to obtain a favorable embedding using unsupervised learning on $\mathcal{U}$, with the resulting embedding used on the few-shot learning task. These baselines are not meta-learning approaches, however, we can train them with the same target task training set as UMTRA. Similar to [14], we compare the following unsupervised pre-training approaches: ACAI [5], BiGAN [10], DeepCluster [6] and InfoGAN [7]. These up-to-date approaches cover a wide range of the recent advances in the area of unsupervised feature learning. Finally, we also compare against the CACTUs unsupervised meta-learning algorithm proposed in the [14], combined with MAML and ProtoNets [30]. As a note, another unsupervised meta-learning approach related to UMTRA and CACTUs is AAL [2]. However, as [2] doesn't compare against stock MAML, the results are not directly comparable.

Table 4, columns three to six, shows the results of the experiments. For the UMTRA approach we trained for 6000 meta-iterations for the 5-way, and 36,000 meta-iterations for the 20-way classifications. Our approach, with the proposed hyperparameter settings outperforms, with large margins, training from scratch and the approaches based on unsupervised representation learning. UMTRA also outperforms, with a smaller margin, the CACTUs approach on all metrics, and in combination with both MAML and ProtoNets.

As expected, the supervised meta-learning baselines perform better than UMTRA. To put this value in perspective, we need to take into consideration the vast difference in the number of labels needed for these approaches. In 5-way one-shot classification, UMTRA obtains a 83.80% accuracy with only 5 labels, while supervised MAML obtains 94.46% but requires 24005 labels. For 5-way 5-shot classification UMTRA obtains a 95.43% accuracy with only 25 labels, while supervised MAML obtains 98.83% with 24025.

## 4.2 UMTRA on the Mini-Imagenet dataset

The Mini-Imagenet dataset was introduced by [27] as a subset of the ImageNet dataset [9], suitable as a benchmark for few-shot learning algorithms. The dataset is limited to 100 classes, each with 600 images. We divide our dataset into train, validation and test subsets according to the experimental protocol proposed by [31]. The classifier network is similar to the one used in [11].

Since Mini-Imagenet is a dataset with larger images and more complex classes compared to Omniglot, we need to choose augmentation functions suitable to the model. We had investigated several simple

choices involving random flips, shifts, rotation, and color changes. In addition to these hand-crafted algorithms, we also investigated the learned auto-augmentation method proposed in [8]. Table 1, right, shows the accuracy results for the tested augmentation functions. We found that auto-augmentation provided the best results, thus this approach was used in the remainder of the experiments.

The last four columns of Table 4 lists the experimental results for few-shot classification learning on the Mini-Imagenet dataset. Similar to the Omniglot dataset, UMTRA performs better than learning from scratch and all the approaches that use unsupervised representation learning. It performs weaker than supervised meta-learning approaches that use labeled data. Compared to the various combinations involving the CACTUs unsupervised meta-learning algorithm, UMTRA performs better on 5-way one-shot classification, while it is outperformed by the CACTUs-MAML with DeepCluster combination for the 5, 20 and 50 shot classification.

A possible question might be raised whether the improvements we see are due to the meta-learning process or due to the augmentation enriching the few shot dataset. To investigate this, we performed several experiments on Omniglot and Mini-Imagenet by training the target tasks from scratch on the augmented target dataset. For 5-way, 1-shot learning on Omniglot the accuracy was: training from scratch 52.5%, training from scratch with augmentation 55.8%, UMTRA 83.8%. For MiniImagenet the numbers were: from scratch without augmentation 27.6%, from scratch with augmentation 28.8%, UMTRA 39.93%. We conclude that while augmentation does provide a (minor) improvement on the target training by itself, the majority of the improvement shown by UMTRA is due to the meta-learning process.

The results on Omniglot and Mini-Imagenet allow us to draw the preliminary conclusions that unsupervised meta-learning approaches like UMTRA and CACTUs, which generate meta tasks $\mathcal{T}_i$ from the unsupervised training data tend to outperform other approaches for a given unsupervised training set $\mathcal{U}$. UMTRA and CACTUs use different, orthogonal approaches for building $\mathcal{T}$. UMTRA uses the statistical likelihood of picking different classes for the training data of $\mathcal{T}_i$ in case of $K = 1$ and large number of classes, and an augmentation function $\mathcal{T}$ for the validation data. CACTUs relies on an unsupervised clustering algorithm to provide a statistical likelihood of difference and sameness in the training and validation data of $\mathcal{T}_i$. Except in the case of UMTRA with $\mathcal{A} = \mathbb{1}$, both approaches require domain specific knowledge. The choice of the right augmentation function for UMTRA, the right clustering approach for CACTUs, and the other hyperparameters (for both approaches) have a strong impact on the performance.

## 5    Conclusions

In this paper, we described the UMTRA algorithm for few-shot and one-shot learning of classifiers. UMTRA performs meta-learning on an **unlabeled dataset** in an unsupervised fashion, without putting any constraint on the classifier network architecture. Experimental studies over the few-shot learning image benchmarks Omniglot and Mini-Imagenet show that UMTRA outperforms learning-from-scratch approaches and approaches based on unsupervised representation learning. It alternated in obtaining by best result with the recently proposed CACTUs algorithm that takes a different approach to unsupervised meta-learning by applying clustering on an unlabeled dataset. The statistical sampling and augmentation performed by UMTRA can be seen as a cheaper alternative to the dataset-wide clustering performed by CACTUs. The results also open the possibility that these approaches might be orthogonal, and in combination might yield an even better performance. For all experiments, UMTRA performed worse than the equivalent supervised meta-learning approach - but requiring 3-4 orders of magnitude less labeled data. The supplemental material shows that UMTRA is not limited to image classification but it can be applied to other tasks as well, such as video classification.

**Acknowledgements:** This research is based upon work supported in parts by the National Science Foundation under Grant numbers IIS-1409823 and IIS-1741431 and Office of the Director of National Intelligence (ODNI), Intelligence Advanced Research Projects Activity (IARPA), via IARPA R&D Contract No. D17PC00345. The views, findings, opinions, and conclusions or recommendations contained herein are those of the authors and should not be interpreted as necessarily representing the official policies or endorsements, either expressed or implied, of the NSF, ODNI, IARPA, or the U.S. Government. The U.S. Government is authorized to reproduce and distribute reprints for Governmental purposes notwithstanding any copyright annotation thereon.

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
