[Supplementary Material · Neurips_UMTRA_SupMat.pdf]

**Supplementary Material for Unsupervised Meta-Learning for Few-Shot Image Classification**

**Evolution of accuracy during training**

In these series of experiments we study the evolution of the accuracy obtained after a specific number of gradient training steps during the target learning phase. The results for Omniglot are shown in Figure 1 (with K=1), while those for Mini-Imagenet in Figure 2 with K values of 1, 5 and 20. For both datasets, we compare learning from scratch, UMTRA and supervised MAML. As expected, both MAML and UMTRA reach their accuracy plateau very quickly during target training, while learning from scratch takes a larger number of training steps. Further training does not appear to provide any benefit for either approach. The results are averaged among 1000 tasks. This demonstrates that UMTRA has the capacity to learn to adapt to novel tasks by just looking at unlabeled data and generating tasks from that dataset in an unsupervised manner. An interesting phenomena happens with $K = 5$ and $K = 20$ values for Mini-Imagenet: the accuracy curve of UMTRA dips after the first iteration, and it takes several iterations to recover. We conjecture that this is a result of the fact that UMTRA sets $K = 1$ during meta-learning, thus the resulting network is best optimized to learn from one sample per class.

Figure 1: The accuracy curves during the target training task on the Omniglot dataset for $K = 1$. The band around lines denotes a $95\%$ confidence interval.

**Feature Representations**

To compare generalization of training from scratch, UMTRA and supervised MAML, we visualize the activations of the last hidden layer of the network on Omniglot dataset by t-SNE. We compare all of the methods on the same target training task which is constructed by sampling five characters from test data and selecting one image from each character class randomly. Each character has 20 different instances. Figure 3 shows the t-SNE visualization of raw pixel values of these 100 images. Instances which are sampled for the one-shot learning task are connected to each other by dotted lines. Figure 4 shows the visualization of the last hidden layer activations for the same task. UMTRA as well as MAML can adapt quickly to a feature space which has a better generalization than training from scratch.

**Video Domain**

In this section, we show how the UMTRA can be applied to video action recognition, a domain significantly more complex and data intensive than the one used in the few-shot learning benchmarks such as Omniglot and Mini-Imagenet. To the best of our knowledge, we are the first to apply meta-learning to video action recognition. We perform our comparisons using one of the standard video action recognition datasets, UCF-101 . UCF-101 includes 101 action classes divided into five types: Human-Object Interaction, Body-Motion Only, Human Human Interaction, Playing Musical Instruments and Sports. The dataset is composed of snippets of Youtube videos. Many videos have poor lighting, cluttered background and severe camera motion. As the classifier on which to apply the meta-learning process, we use a 3D convolution network, C3D .

Performing unsupervised meta-learning on video data, requires several adjustments to the UMTRA workflow, with regards to the initialization of the classifier, the split between meta-learning data and testing data, and the augmentation function.

Figure 2: The accuracy curves during the target training task on the Mini-Imagenet dataset. Accuracy curves are shown for $K = 1$ (Top left), $K = 5$ (Top right), and $K = 20$ (Bottom). The band around lines denotes a 95% confidence interval.

Figure 3: t-SNE on the Omniglot raw pixel values.

First, networks of the complexity of C3D cannot be learned from scratch using the limited amount of data available in few-shot learning. In the video action recognition research, it is common practice to start with a network that had been pre-trained on a large dataset, such as Sports-1M dataset , an approach we also use in all our experiments.

Second, we make the choice to use two different datasets for the meta-learning phase (Kinetics) and for the few-shot learning and evaluation (UCF-101 ). This gives us a larger dataset for training since Kinetics contains 400 actions, but it introduces an additional challenge of domain-shift: the network is pre-trained on Sports-1M, meta-trained on Kinetics and few-shot trained on UCF-101. This approach, however, closely resembles the practical setup when we need to do few-shot learning on a novel domain. When using the Kinetics dataset, we limit it to 20 instances per class.

For the augmentation function $\mathcal{A}$, working in the video domain opens a new possibility, of creating an augmented sample by choosing a temporally shifted video fragment from the same video. In other words, we can use self supervision in video domain: The augmentation is to sample another part of the same video clip. Figure 5 shows some samples of these augmentations. In our experiments, we have experimented both with UMTRA (using a Kinetics dataset stripped from labels), and supervised meta-learning (retaining the labels on Kinetics). This supervised meta-learning experiment is also significant because, to the best of our knowledge, meta-learning has never been applied to human action recognition from videos.

Training from Scratch

UMTRA

MAML

Figure 4: Visualization of the last hidden layer activation values by t-SNE on the Omniglot dataset before target task training (Left), and after target task training (Right). Visualized features are shown for training from scratch (Top), UMTRA (Middle), and MAML (Bottom). Each class is shown by a different color and shape. From each class one instance is used for target task training. Training instances are denoted by larger and lighter symbols and are connected to each other by dotted lines

Figure 5: Example of the training data and the augmentation function $\mathcal{A}$ for video. The training data $x$ is a 16 frame segment starting from a random time in the video sample (Here we show three frames of a sample at each column). The validation data $x' = \mathcal{A}(x)$ is also a 16 frame segment, starting from a different, randomly selected time *from the same video sample*.

Table 1: Accuracy and F1-Score for a 5-way, one-shot classifier trained and evaluated on classes sampled from UCF-101. All training (even for "training from scratch"), employ a C3D network pre-trained on Sports-1M. For all approaches, none of the UCF-101 classes was seen during pre- or meta-learning.

| Algorithm | Test Accuracy / F1-Score |
|---|---|
| *Training from scratch* | 29.30 / 20.48 |
| Pre-trained on Kinetics | 45.51 / 42.49 |
| UMTRA on unlabeled Kinetics (ours) | **60.33 / 58.47** |
| Supervised MAML on Kinetics | 71.08 / 69.44 |

In our evaluation, we perform 30 different experiments. At each experiment we sample 5 classes from UCF-101, perform the one-shot learning, and evaluate the classifier on all the examples for the 5 classes from UCF-101. As the number of samples per class are not the same for all classes, in Table 1 we report both the accuracy and F1-score.

The results allow us to draw several conclusions. The relative accuracy ranking between training from scratch, pre-training and unsupervised meta-learning and supervised meta-learning remained unchanged. Supervised meta-learning had proven feasible for one-shot classifier training for video action recognition. UMTRA performs better than other approaches that use unsupervised data. Finally, we found that the domain shift from Kinetics to UCF-101 was successful.