[Reviews · NeurIPS 2019]

Reviewer 1



---------After Rebuttal--------- After reading through the rebuttal and considering the other reviewer's comments, I have decided to raise my score to lean towards acceptance. In the rebuttal, the authors have run additional experiments demonstrating that their method does outperform augmentation pre-training baselines by a reasonably large margin. However, I still think the reliance on these explicitly defined set of transformations leaves something to be desired. Also, I don't quite understand the last point of their rebuttal "for videos we do not do augmentation" when Figure 5 in their supplementary says "Example of the training data and the augmentation function A for video." ---------Before Rebuttal--------- The ideas presented in this work are clear and the algorithm is succinct. For the proposed method, one concern is the actual complexity of the components involved in meta-training. The approach of using Auto-Augment within UMTRA would suggest significant computational overhead for a new dataset. Also, the number of potential hyperparameters in this extension of MAML in addition to the original hyperparameters introduced by MAML would make this method unwieldy to use in practice. Both these issues reduce the practicality of this idea and harm the significance. Also, the overall originality is somewhat limited as it is a fairly direct combination of data augmentation with meta-learning.

Reviewer 2



The notion of combining unsupervised learning with supervised learning has a long history, but this paper takes an interesting viewpoint of assuming that randomly drawn images come from different classes. Of course, the validity of this assumption is highly dependent on the number of underlying classes, but their equation 1 illustrates the rough scaling. They then use data augmentation to ensure that similar images are assigned appropriately. I found the work to be clear and well reasoned. The most significant contribution would be the assumption that all samples are in a different class. This assumption is also a very strong assumption -- and they do not explain how a person might test this assumption on any particular problem where classes are not previously articulated (as they are in imagenet). This makes the generalizability of this work uncertain. I am not concerned about the relative performance as much as having the authors put more detail into justifying why this would work on data collected from, say, autonomous vehicles or social media or commercial satellites. How would we estimate if "N << C"?

Reviewer 3



======= AFTER REBUTTAL ========= I will raise my score after the rebuttal. I feel that the new dataset, the video results and the precise responses really helped with the clarity. ======================== Abstract: -> I’d recommend against putting specific method names and comparisons into the abstract because this makes the abstract not age very well as methods change and improve. It also makes the contribution seem fairly incremental, rather than providing a new insight. Introduction: -> I would recommend against calling these auxiliary tasks. The meta-training tasks are not what people would typically think of when talking about auxiliary objectives. -> minor: “same supervised learning format”. Not clear what this means, maybe reword? -> As a reader, mentioning that 2 other methods did the same thing recently doesn’t give me any additional insight in the introduction and doesn’t really add value there. It should be discussed in the related work and perhaps in the method section, but making it so prevalent in the intro and the abstract aren’t adding significant value in your contribution. -> I think a more detailed discussion of why the right comparison points are unsupervised representation learning algorithms versus other things is going to be helpful. Related Work: “We need to point out however, that the meta-learning phase uses custom algorithms in these approaches as well” -> I don’t understand. I thought the whole point was to not use custom algorithms, and you say so yourself. Why the contradiction? -> You mention CACTus and AAL. Can you discuss detailed comparisons and relationships in the related work? -> Related work which was missed from RL: https://arxiv.org/abs/1806.04640 -> The meta-testing performance is going to critically depend on the overlap between the imaginary task distribution and the testing distribution. Would help with clarity if this was mentioned early in the paper. Method: “with samples drawn from the same distribution as the target task” -> Would help to formalize this? What does it mean for this to be drawn from the same distribution. Is there a distribution p(x) on images x? -> Minor: What is $N$ in the method? -> Minor: What is $K$, why do we need $NxK$. These should be clarified with a preliminaries section. “The exact labels used during the meta-training step are not relevant, as they are discarded during the meta-training phase.” -> Minor: This line should be reworded to be more clear. -> A preliminaries section would be really useful to clarify the notation. For instance, the term "target learning phase” is used in many places but not clearly defined. -> I think the choice of augmentation function might be more than just a hyperparameter. It seems very crucial to the performance of the algorithm. For instance, if we were classifying boats, just rotating or translating may not give us the validation we want in general. We may need something that looks significantly more different, for instance a different boat altogether. Can we potentially provide any guarantees on generalization here when the augmentation function is very domain specific? -> The rationale in 170 seems like a bit confusing to me. Why is it unbiased? Could the authors provide a bit more clarity? Experiments: I liked the control hyperparameter setting that the authors use for UMTRA evaluation. The additional comparison of the effect of UMTRA augmentation range on the performance is pretty interesting. The improvement on omniglot is pretty striking, and the relatively much smaller improvement on mini imagenet suggests that perhaps the augmentation in one is much easier to do effectively than the other. I think the overall experiments section is pretty strong, with impressive results on omniglot, and weaker results on mini-imagenet. I think the algorithm is in many ways somewhat simplistic, and is very strongly dependent on the choice of augmentation. This can be pretty nontrivial for more challenging domains, and a more general discussion of this would be important, as well as perhaps results on another domain to showcase the generality of the augmentation method across domains.

Reviewer 4



Post-rebuttal: ---------- While I appreciate that the authors have emphasized the importance of domain knowledge for constructing augmentation functions, I believe it is essential that they demonstrate the effectiveness of the method on other (non-vision) problems. The method, as it is currently presented, is probably better suited to a computer vision venue as opposed to a general machine learning venue. Pre-rebuttal: ---------- Major Comments ---------- My primary concern with the UMTRA framework is the hand-designed augmentation function. My interpretation of the data generation problem is as follows: Given one (or more, from the same class) data points, x_{1:n}, we would like to generate data point x_{n+1} (or possibly multiple data points) that have the same class with high probability. The approach to this problem taken in CACTUs is to perform this class-conditioned unsupervised learning problem by constructing a representation of the data in which clustering tools can be used effectively. In contrast to the automatic clustering approach, UMTRA effectively aims to hand-solve this class-conditional unsupervised learning problem. I worry that this is quite a fragile approach to this problem. Moreover, the authors choose to demonstrate their approach on data for which rough approximations of this unsupervised problem are easily solved. In particular, the authors leverage invariances that are (mostly) unique to image/video data to generate samples (for example, flipping an image). I am skeptical that this approach can be easily performed for other domains. Consider two similar, common machine learning problems: credit card fraud detection and spam detection. How could one hand-augment or perturb samples x_i to maintain their status as non-spam or spam? While it is not impossible that the approach presented herein may be valid for general problems by leveraging domain-specific knowledge, I am skeptical that it is generally applicable. The degraded performance of Mini-Imagenet also reflects these fears. I would be willing to improve my numerical evaluation of this work if the authors could show that their appoach is capable of effectively handling problems without obvious invariances. Overall, I found the paper to be fairly clearly written. The authors could improve clarify by being more explicit about the generating distributions of each quantity introduced. Minor Comments ---------- - I believe in line 168, epsilon should have variance sigma^2? - It is unclear how much it impacts training if samples from the same class appear multiple times in the few-shot learning problem. While the authors give a reasonable statistical argument that this is infrequent, an experiment showing the impact of this would be interesting.

[Author Response · NeurIPS 2019]

We thank the reviewers for the time and expertise they have invested in these reviews. We appreciate positive comments about the paper like connecting the existing meta-learning frameworks with unsupervised/self-supervised feature learning frameworks, clear presentation of the ideas, good control parameter experiments, and novel approach to the unsupervised meta-learning problem. Furthermore, the insightful comments we received point us towards several directions in which this work can be extended in the future. We focus on answering to the posed questions by the reviewers and provide feedback related to improvements suggestions.

Answers to reviewer #1.

**How much does hand-crafted knowledge play a role in the performance of the proposed method?** Domain knowledge, such as knowing what transformations of the image retain the classification, does play a role in the performance of the augmentation function. See left side of Table 1 where various hand-crafted augmentations performed at accuracy of 30.16%, 32.80%, 35.09% against the 24.17% baseline. It turned out that the learned auto-augment outperformed all the hand-crafted methods with 39.93%.

**In particular, a simple baseline would be to perform few-shot training on models trained with the augmentation methods proposed. That seems like a more informative/reasonable baseline than training from scratch.** We agree, we implemented this, and here are the results. For 5-way, 1-shot learning on Omniglot the accuracy was: training from scratch 52.5%, training from scratch with augmentation 55.8%, UMTRA 83.8%. For MiniImagenet the numbers were: from scratch without augmentation 27.6%, from scratch with augmentation 28.8%, UMTRA 39.93%.

**The impact of this work could be more substantial if the proposed method were generalized beyond data-augmentation to unsupervised learning in the few-shot setting using other proposed self-supervised techniques.** While the main body of the paper focuses on augmentation techniques, its application to the video domain in the supplemental material is using self-supervision: we randomly select 16 frames from one video for training and another 16 frames from the same video for validation. For results, see table at supplemental material page 4.

**The approach of using Auto-Augment within UMTRA would suggest significant computational overhead for a new dataset:** This overhead indeed exists during the meta-training time. Note however, that this is an offline process which needs to be done once per domain. No augmentation is done during the target learning phase.

**How would augmentation work for non-image data?** The choice of augmentation is domain specific. For example, in our supplemental material, for videos we do not do augmentation and just pick two different parts of the video. Look at the Figure 5 at supplemental material.

Answers to reviewer #3

**that all samples are in a different class (...) how a person might test this assumption on any particular problem where classes are not previously articulated (as they are in imagenet). (...) How would we estimate if "N $\ll$ C"?. (...) Does one need to fully articulate the task space – how do you project all possible arrangements of the data into separate classes down into some effective number of classes, c?** Indeed, for the unlabeled datasets that we use for meta-learning, we do not know the exact value of $C$. This is ok, because UMTRA does not take $C$ as a parameter. We only need a rough estimate of it to ensure that "N $\ll$ C" holds, we do not need to fully articulate the task space. For instance, if we download 10,000 random videos from YouTube, we can estimate that there will be at least hundreds of different activities in them.

Answers to reviewer #6

| Algorithm (N, K) | (5, 1) | (5, 5) | (5, 10) |
|---|---|---|---|
| *Training from scratch* | 26.86 | 39.65 | 50.61 |
| UMTRA | **33.43** | **50.19** | **58.84** |

**I would run experiments on one other domain to assure readers of the generality of the system across domains with widely varying appearances within the dataset.** As requested, we run UMTRA on the CelebA dataset (which is unbalanced in terms of the number of examples in each class) for identity recognition and obtained the results shown in table above. Given 5 new identities and one image of each, UMTRA is able to learn the task better than training from scratch. The supplemental material also contains results on another domain, video classification.

Answers to reviewer #7

**The most important further contribution would be demonstrating that the augmentation approach can either be effectively automated in all cases, or that a rough hand-designed augmentation approach can be found that works in other domains.** We believe that some degree of domain knowledge is a necessary input into the design of the augmentation algorithm. For instance, we know that photos are crops of 2D projections of 3D scenes, which imply certain invariances that can be exploited by the augmentation. This augmentation can be then used across all scenarios involving photos. For instance, the CelebA results above were obtained with the same augmentation used for MiniImagenet.

[Meta-Review · NeurIPS 2019]

This paper is extremely borderline, and the reviewers were split during the discussion. The video classification experiment in the appendix is quite nice and is critical for illustrating the generality of the method beyond image classification. However, these results appear somewhat preliminary with no comparisons to unsupervised learning methods, and should be highlighted in the main text of the paper, rather than coming across as an after thought. Further examples of results in other domains would further strengthen the contribution of this paper to the general ML community. As it stands, we think that the contributions of the paper are valuable to the NeurIPS community. In the final version, we strongly encourage the authors to include a more thorough discussion of the broader method, and include results in the main paper that better exemplify and analyze the generality of the method.